# *Francisella tularensis* Subspecies *holarctica* and Tularemia in Germany

**DOI:** 10.3390/microorganisms8091448

**Published:** 2020-09-22

**Authors:** Sandra Appelt, Mirko Faber, Kristin Köppen, Daniela Jacob, Roland Grunow, Klaus Heuner

**Affiliations:** 1Centre for Biological Threats and Special Pathogens (ZBS 2), Robert Koch Institute, 13353 Berlin, Germany; appelts@rki.de (S.A.); jacobd@rki.de (D.J.); grunowro@rki.de (R.G.); 2Gastrointestinal Infections, Zoonoses and Tropical Infections (Division 35), Department for Infectious Disease Epidemiology, Robert Koch Institute, 13353 Berlin, Germany; faberm@rki.de; 3Cellular Interactions of Bacterial Pathogens, ZBS 2, Robert Koch Institute, 13353 Berlin, Germany; koeppenk@rki.de

**Keywords:** *Francisella tularensis* subspecies *holarctica*, W12-1067, tularemia, rabbit fever, zoonotic disease, Germany

## Abstract

Tularemia is a zoonotic disease caused by *Francisella tularensis* a small, pleomorphic, facultative intracellular bacterium. In Europe, infections in animals and humans are caused mainly by *Francisella tularensis* subspecies *holarctica*. Humans can be exposed to the pathogen directly and indirectly through contact with sick animals, carcasses, mosquitoes and ticks, environmental sources such as contaminated water or soil, and food. So far, *F. tularensis* subsp. *holarctica* is the only *Francisella* species known to cause tularemia in Germany. On the basis of surveillance data, outbreak investigations, and literature, we review herein the epidemiological situation—noteworthy clinical cases next to genetic diversity of *F. tularensis* subsp. *holarctica* strains isolated from patients. In the last 15 years, the yearly number of notified cases of tularemia has increased steadily in Germany, suggesting that the disease is re-emerging. By sequencing *F. tularensis* subsp. *holarctica* genomes, knowledge has been added to recent findings, completing the picture of genotypic diversity and geographical segregation of *Francisella* clades in Germany. Here, we also shortly summarize the current knowledge about a new *Francisella* species (*Francisella* sp. strain W12-1067) that has been recently identified in Germany. This species is the second *Francisella* species discovered in Germany.

## 1. Introduction

Tularemia, also called “rabbit fever”, is a rare but potentially severe zoonosis caused by *Francisella tularensis*. The pleomorphic bacterium is non-motile and non-sporulating, and proliferates efficiently in different host cells, in vivo mainly in macrophages [1,2,3]. The pathogen has a wide host range including mammals, birds, amphibians, fishes, and invertebrates [4,5]. Transmission to humans occurs through contact with sick animals, primarily free-living lagomorphs (hares and rabbits), as well as animal carcasses, and moreover through arthropod vectors (mosquitoes and ticks), environmental sources (water, dust, aerosol, and soil), and sometimes food and drinking water [5,6,7,8,9]. Thereby, hunters are a typical risk group exposed through handling (skinning, preparing, or consuming) meat of infected animals [10,11,12,13]. Human to human transmission has not been described thus far, despite a case of tularemia after organ transplantation [14].

The species *Francisella tularensis* comprises four main subspecies (*Francisella tularensis* subsp. *tularensis*, *F. tularensis* subsp. *holarctica*, *F. tularensis* subsp. *mediasiatica*, and *F. tularensis* subsp. *novicida*), with distinct geographical distributions and high to low pathogenicity for humans. Two subspecies are of clinical relevance: *F. tularensis* subsp. *tularensis* and *F. tularensis* subsp. *holarctica*. *F. tularensis* subsp. *tularensis* is predominantly found in North America while *F. tularensis* subsp. *holarctica* is prevalent in the whole northern hemisphere. Recent reports of tularemia in Australia caused by *F. tularensis* subsp. *holarctica* demonstrated this subspecies to be present also in the Southern hemisphere [15,16]. *F. tularensis* subsp. *holarctica* seems to originate from North America or Asia [17,18,19,20] and has spread through Scandinavia to southern parts of Europe, passing German territories as early as the 19th century, as described in historical records and more recent investigations [17,18,21]. Today, *F. tularensis* subsp. *holarctica* shows a broad genetic variety in Europe. Two clades are most abundant: the basal clade B.6 (biovar I, erythromycin-sensitive) is more prominent in Western Europe, and basal clade B.12 (biovar II, erythromycin-resistant) is more prevalent in north-eastern Europe [4,5,17,19,21,22,23,24,25,26,27,28,29]. Indeed, evidence for a geographical north to south segregation of both basal clades was recently demonstrated for Germany as well, and new clades as well as a new *Francisella* species (*Francisella* sp. isolate W12-1067) were identified [22,30]. Interestingly, this isolate is the only *Francisella* species described as being present in Germany next to *F. tularensis* subsp. *holarctica* [21,30]. This species seems not to be clinically relevant regarding the yet available knowledge, and is described in Section 4.

In Germany, tularemia is a notifiable disease. Important epidemiological aspects, such as clinical aspects and diagnosis, seasonality, outbreaks, and putative natural reservoirs, are discussed in Section 3 of this review. While today tularemia is rarely diagnosed in Germany, a 10-fold increase in notified cases during the last 15 years indicates a re-emergence of the pathogen [21].

This review summarizes our current knowledge about tularemia in Germany, which might also be of interest to other European countries with a similar epidemiological situation. Included in the review are data of sequenced genomes of *Francisella* isolates from patients recovered in 2019, adding new findings about genetic diversity and the geographical segregation of clades in Germany.

## 2. *Francisella tularensis* Subsp. *holarctica*—Genetic Diversity and Geographic Distribution

A clonal structure of the pathogen is considered. For epidemiological studies of tularemia, canonical single-nucleotide polymorphisms (canSNPs) at whole-genome scale are used to genotype *Francisella tularensis* subsp. *holarctica* strains [31,32,33]. Thus far, four major clades have been described: B.4, B.6, B.12, and B.16 [18,19]. These clades can be divided further into subclades. Clade B.4 is very rarely found in Germany and B.16 can be found mainly in Japan (biovar *japonica*). The clades B.6 and B.12 are most abundant in Europe and a spatial segregation between western and north-eastern parts in Europe has been described [4,5,17,19,21,22,23,24,25,26,27,28,29].

Although a high genetic diversity has been reported from Scandinavia, it has been hypothesized that *F. tularensis* subsp. *holarctica* strains in Western Europe have expanded from closely related strains with a long-range dispersal over time, slow replication, and long-term persistence in the environment [17,18,31,34]. Interestingly, a high diversity of different *F. tularensis* subsp. *holarctica* strains could also be found in Germany, especially in northern parts [21,35,36]. Additionally, recent investigations identified various different B. 6 (B.51, B.50, B52, B.63) and B. 12 (B.34, B.36, B. 71, B. 73, B. 74, B.77) subclades to be present in Germany. Moreover, by sequencing 12 *F. tularensis* subsp. *holarctica* isolates recovered from clinical samples in 2019, we identified two additional subclades: B.12/B.80 and B.6/B90 (see Figure 1), which have not been previously isolated from patients in Germany [22]. Of these 12 isolates, nine clustered into clade B.6 and three into clade B.12. Thus, the ratio of the number of strains belonging to biovar I or II was found to be in agreement with other investigations studying *F. tularensis* subsp. *holarctica* diversity in Germany [22,37,38,39]. The genetic diversity of *F. tularensis* subsp. *holarctica* in Germany, as in other European countries, seems to be still underestimated and further investigations are needed, as shown by recent data obtained in France (82 new B.6 subclades were described) [40]. However, the definition of further “sub-sub-sub”-clades (now B. > 300) should be a subject of discussion, since sometimes only very few SNPs are different between strains of such subclades, and a further subdivision of isolates may not be reasonable.

Next to high diversity of *F. tularensis* subsp. *holarctica* strains in Germany, recent investigation could show the spatial segregation of clades (B.6 and B.12 clade) within the country [22]. It was found that B.6 clade members are more abundant in the southwestern parts of Germany whereas B.12 clade members are to be found more often in the northeast [22]. These finding could be supported by new acquired data of *F. tularensis* subsp. *holarctica* isolates investigated herein. The identified B.6 clade members were recovered from patient samples collected in southwestern parts of Germany and B.12 clade members from samples collected in northeastern parts. Indeed, these findings fit into the picture of spatial segregation of clades within western and north-eastern parts of Europe as mentioned above. Thus, taking into account latest findings, Germany might be a “melting pot” for the species, a region where strains become mixed and new genetic variants arise [21,22,36,41,42].

## 3. Tularemia in Germany

### 3.1. Epidemiology of Notified Cases

Tularemia is a notifiable disease according to Germany’s infection protection act of 2001. The current surveillance case definition applies to persons with clinical symptoms compatible with tularemia together with one of the following four laboratory criteria: antigen detection by, e.g., enzyme immune assays or immunofluorescence assays; isolation from bacterial culture; detection of specific nucleic acids, e.g., by polymerase chain reaction; or detection of specific antibodies (titer increase in consecutive serum samples or one clearly elevated titer (https://www.rki.de/DE/Content/Infekt/IfSG/Falldefinition/Downloads/Falldefinitionen_des_RKI_2019.pdf)). Tularemia is likely subject to significant underdiagnoses and underreporting in Germany as the disease is generally rare, and, in patients presenting with lymphadenitis and fever, tularemia is rarely considered as a differential diagnosis by clinicians and diagnostic laboratories.

Between 1 January 2002 and 31 December 2019, 435 cases of tularemia were notified in Germany (Figure 2), corresponding to a mean yearly incidence of 0.29 cases per million population (range: 0.01–0.88). Of the 435 cases reported, 387 were sporadic cases and 48 belonged to a cluster of cases. Only for 39 of 435 cases (9.0%) was a relevant travel history in the likely period of infection notified; thus, most infections were likely acquired in Germany. Age of the patients ranged from 1 to 99 years (mean: 47.5) and the male to female ratio was 2.11. While a median of three annual cases were notified from 2002 through 2006, an overall increase from 2007 through 2012 and a year on year increase between 2013 (*n* = 20) and 2019 (*n* = 72) was observed (Figure 2). Thus, the incidence of reported cases is on a level comparable to the yearly mean in Germany in the 1950s when, likely due to the socio-economic situation after the war, cases had surged. It is unclear whether the increase in the last 15 years is due to an actual increase in infection pressure and clinical cases or whether it is the result of increased awareness and more frequent testing. However, a relatively stable proportion of hospitalized cases suggest that the increase is not the result of a change in sensitivity of the surveillance system.

Tularemia is a seasonal disease in Germany, with most patients (63%) reporting symptom onset from July through November (Figure 3) when reservoir animal populations are peaking and frequent outdoor activities (such as hunting, farming, fishing, hiking, etc.) facilitate contact between wildlife and humans. This is in concordance with the seasonal occurrence of tularemia cases in Europe [20,21]. Imported cases only account for a small fraction of the total case load (39 of 360, 11.3%), peaking after the summer holiday season in August and after Christmas/New Year.

Cases are reported from almost all Federal States of Germany. Between 2002 and 2019, the highest mean annual incidences were recorded in parts of Saxony-Anhalt, Baden-Württemberg, and Brandenburg (Figure 4). Although there is large variability in the regional incidence of tularemia, long-term surveillance data indicate that the pathogen can be found all over Germany. The variation in the number of reported cases from the different Federal States could be explained by (i) variations in actual disease incidence as a result of varying exposure risks or infection pressure or (ii) variations in diagnostic consideration of tularemia and reporting activity due to differences in awareness of healthcare workers for the disease. Comparing reported cases and results of serological studies indicate that only a fraction of infections are diagnosed and reported [10,41,42,43,44,45]. Two cross-sectional studies have shown a relatively high seroprevalence: one population-representative study from 2004 with 6617 sera and one study conducted in a small town in Baden-Württemberg in 2009 with 2416 sera, which revealed positive results in 0.23% and 2.3% of the sera, respectively [46,47]. Serological studies in hunters as a putative population at high risk have shown a seroprevalence of up to 1.7% [10].

### 3.2. Outbreaks of Tularemia in Humans

Outbreaks or clusters of tularemia (defined as at least two cases with an epidemiological link, such as a common source of exposure) are apparently rare in Germany. Between 2002 and 2019, 14 case clusters were reported. Except for two with a connection to situations in other countries and one large outbreak caused by grape must (unintentional contamination) [7,8,9], the remaining 11 clusters consisted of 2 to 10 cases and were associated with contact with wild animals, nine of them in the context of hunting activities [12,13,37,48,49,50,51,52]. Two outbreaks are presented in more detail below.

Eight of the above clusters were reported in connection with consumption of or contact with infected hares. The largest occurrence was in 2005, with a total of 10 affected hunters participating in a hare hunting event [12,13]. Indeed, prevalence studies in ticks in the southwest of Germany revealed the presence of *Francisella* in 8% of 916 investigated *Ixodes Ricinus*, while *Dermacentor* species clustered with *Francisella* endosymbionts [41]. For cases that were not part of clusters, contact with hares, rabbits, wild boars, and deer, and also tick bites and mosquitoes, have been suspected as the source of infection [37,52,53,54,55,56]. Some wild animals (raccoon dog, red fox, wild boar, hares, voles) have been suggested as serving as natural reservoirs for *F. tularensis* subsp. *holarctica* or as sentinels for tularemia [36,38,41,43,44,47,54,57,58,59,60,61]. The role of ticks and mosquitoes for the transmission of tularemia in Europe likely differs between countries and is still under investigation [22,26,37,39,41,55,62,63,64,65].

Outbreak 1: An uncommon outbreak of oropharyngeal tularemia occurred in 2016 in Rhineland-Palatinate after the consumption of freshly pressed grape must during a grape harvest [7,8,9]. Among 29 harvesters, six developed clinical symptoms compatible with tularemia (swollen cervical lymph nodes, fever, chills, and diarrhea) 4 to 8 days after the exposure. Tularemia was proven serologically in all patients and they required an antibiotic treatment duration of more than 14 days [9]. The must served to the participants of the harvest was collected by a mechanical harvester and pressed at the winery. Six weeks after the event, the contaminated must-derived products (sweet reserve (SR) and young wine (YW)) were analyzed for the presence of *F. tularensis*, *F. tularensis* chromosomal DNA, and DNA of the putative source (vector) of contamination [8]. No bacterial isolate could be obtained, but the YW contained the amount of 17,000 *Francisella* genome equivalents per milliliter. A nearly complete draft genome could be generated by next-generation sequencing (NGS) analysis from the DNA isolated from the YW. The genome of the *F. tularensis* subsp. *holarctica* strain contaminating the grape must belongs to the phylogenetic clade B.12/B.34 (see Section 2 and Figure 1), which could be corroborated also by NGS analysis of isolated DNA from an aspirate lymph node sample of one of the patients of this outbreak. In search of the vector responsible for the contamination with *F. tularensis* subsp. *holarctica*, vertebrate-specific cytochrome b sequences could be identified within the isolated DNA from the SR and YW. The revealed cytochrom b sequence analysis identified the putative vector to be *Apodemus sylvaticus* (wood mouse), suggesting that a wood mouse infected with *F. tularensis* subsp. *holarctica* was the source of contamination of the grape must [8]. In conclusion, in this uncommon case, it was proposed that a wood mouse infected with *F. tularensis* subsp. *holarctica* was “collected” by the automatic mechanical harvester, then transferred into the mash car and then into the press, thus contaminating the press and finally 730 L of fresh grape must. This must was then served to the harvester and six people became infected by *F. tularensis* subsp. *holarctica*. It was suggested that rodent control should be implemented in the wine production steps and that freshly pressed must for tasting should be produced generally from hand-picked wine grapes.

Outbreak 2: This occurred in 2018 in a group of hunters in Bavaria. Several hunting dogs and 39 persons were exposed to at least one infected hare. From one of the hares, a *F. tularensis* subsp. *holarctica* strain (A-1338-1-2018) was isolated, belonging to the phylogenetic clade B.12/B.33 (Figure 1 and reference [22]). Thus, the hare could be confirmed as the putative source of infection. In total, 11 of the 39 exposed persons (attack rate 28.2%) developed acute tularemia laboratory confirmed by the detection of specific antibodies. In nine of these patients, the antibody and cytokine response could be monitored over time (Jacob et al., submitted). All samples from hunting dogs, investigated using PCR and cultivation, remained negative for *F. tularensis* subsp. *holarctica*.

### 3.3. Clinical Aspects and Diagnosis

The clinical manifestation of tularemia depends on the entry route of the bacterium into the organism and is defined by ulceroglandular or glandular form, oropharyngeal form, ocularglandular form, respiratory form, and typhoidal form (WHO Guidelines on Tularemia, 2007). The primary common symptoms are fever and enlarged lymph nodes. The incubation period is typically 3 to 5 days with a range of 1 to 14 days, depending on the infectious dose, route of entry, and virulence of the strain. Pulmonary infections with *F. tularensis* subsp. *tularensis* can have a case fatality rate of 30–60%. In the case of complications such as suppuration, pneumonia, and meningitis, convalescence is often prolonged. *F. tularensis* subsp. *holarctica*, the subspecies relevant for human infections in Germany, usually causes a relatively mild form of tularemia in humans. Even without antimicrobial treatment, fatal courses of disease are rare.

Among notified cases in Germany, the most frequent clinical presentations were glandular and ulceroglandular tularemia (Table 1). A total of 12% of the patients presented with mixed forms and 18% could not be assigned (they typically only presented with fever (as well as symptoms less typical for tularemia)). The latter could also represent typhoidal tularemia. Not all authors differentiate between “intestinal” and “oropharyngeal” forms of tularemia (WHO Guidelines on Tularemia, 2007). When symptoms of both forms were present, these are listed under “combination” in Table 1. Collecting additional clinical details during routine surveillance could be considered to allow for a more accurate classification of cases.

Yet, the diagnostic is performed often by detection of specific serum antibodies and/or using isolated DNA for specific qPCRs or PCR assays [32,33,66,67,68,69,70,71,72,73,74,75,76,77]. Cultivation is poorly successful due to the reason that antibiotic treatment of the patient has often already started and therefore seldom leads to positive results [21]. Of the 435 cases of tularemia reported in 2002–2019, 55 (12.6%) were confirmed using an antigen assay, 288 (66.2%) serologically, 68 (15.6%) by culture, and 96 (22.1%) by PCR (some cases were confirmed by a combination of methods). Median time from the onset of symptoms until notification (which typically occurs within two days of diagnosis) was 30 days (inter quartile range: 19–54 days), indicating that diagnosis is often delayed.

## 4. *Francisella* sp. Strain W12-1067 (F-W12)—What Is Known

The new *Francisella* species F-W12 was incidentally identified in Germany in 2012 while screening for the presence of *Legionella* species in a water reservoir of a hospital cooling tower in the framework of an investigation into a cluster of Legionnaires’ disease cases [30]. Further analysis of the obtained bacterial isolate revealed a new *Francisella* species: *Francisella* sp. strain W12-1067. The species is the second identified *Francisella* species in Germany next to *F. tularensis* (subsp. *holarctica*). F-W12 is a close relative of *Allofrancisella guangzhouensis* (formerly named *Francisella guangzhouensis*), which was isolated from an air-conditioning system in China [78,79]. Various homologs of virulence factors of the genus *Francisella* were identified using in silico analysis of the genome sequence of F-W12. Surprisingly, these investigations could show that all genes found on the *Francisella* pathogenicity island (FPI) are missing, but a putative alternative type 6 secretion system is present [30]. Moreover, the strain can persist in co-culture with human cell lines and with amoebae (e.g., *Acanthamoeba lenticulata*) [30,80]. Interestingly, experimental screening assays aiming to elucidate fitness and virulence factors of F-W12 by amoebae co-culture lead to the identification of various known virulence factors of the genus *Francisella*. The majority of the identified genes encode proteins involved in the synthesis or maintenance of the cell envelope (LPS, outer membrane, capsule), starvation (stringent response), or in the metabolism (glycolysis, gluconeogenesis, pentose phosphate pathway). Altogether, the results indicated that F-W12 may be able to replicate in host cells, although natural host cells have not been identified yet [30,80].

To further characterize this new species, we analyzed its core metabolic pathways by applying isotopologue profiling, indicating the presence of a bipartite metabolism of amino acids, glucose, and glycerol. In addition, a myo-inositol (MI) metabolizing gene cluster was identified and it could be demonstrated that F-W12 is able to metabolize MI. F-W12 and also *F. novicida* strain Fx1 are able to use MI as an alternative growth substrate in the absence of glucose [81]. Indeed, the metabolism of F-W12 seems to be more related to the aquatic habitat-associated species *F. novicida* rather than to *F. tularensis* subsp. *holarctica*. Both results support a previously proposed natural water-associated habitat of F-W12 [81].

Further experiments are necessary to investigate the new species’ pathogenicity for protozoa, animals, or humans. This would be a fruitful undertaking, as it could be shown that F-W12 is genetically treatable (generation of mutant strains by a transposon or of specific mutants by site-specific recombination after natural transformation; transformation of plasmids by electroporation). Recently, a new phage integration vector (pFIV-Val) was built, which can be used for integrating genes into the genome of F-W12, as well as for the complementation of specific mutant strains [80,81,82].

## 5. Conclusions

In Germany, tularemia likely represents a re-emerging disease with a high proportion of undiagnosed cases. The increase of cases in the last 15 years may be related to more frequent outdoor activities and contact with wildlife, or to changes in the abundance of *F. tularensis* subsp. *holarctica* in reservoir animals and vectors [65,83]. Increasing awareness and knowledge of the disease among healthcare personnel may facilitate a more timely diagnosis and treatment of cases. More research is needed to be able to assess the burden of disease and to better understand risk factors and routes of infection of tularemia in Germany.

Taking into account the latest findings of the genetic diversity of *F. tularensis* subsp. *holarctica* isolates, Germany might be a “melting pot” for the species, a region where strains get mixed and new genetic variants arise. The identification of the new *Francisella* species F-W12 in Germany indicated that it could make sense to have an “open view” (e.g., methods such as next generation sequencing) when analyzing isolates or probes of patients suffering of tularemia to be able to identify further potentially existing (new) *Francisella* species in Germany. Further experiments are necessary to investigate the pathogenicity of F-W12 for protozoa, animals, and humans, as well as to investigate if this new species may be distributed all over Germany.

## Figures and Tables

**Figure 1 microorganisms-08-01448-f001:**
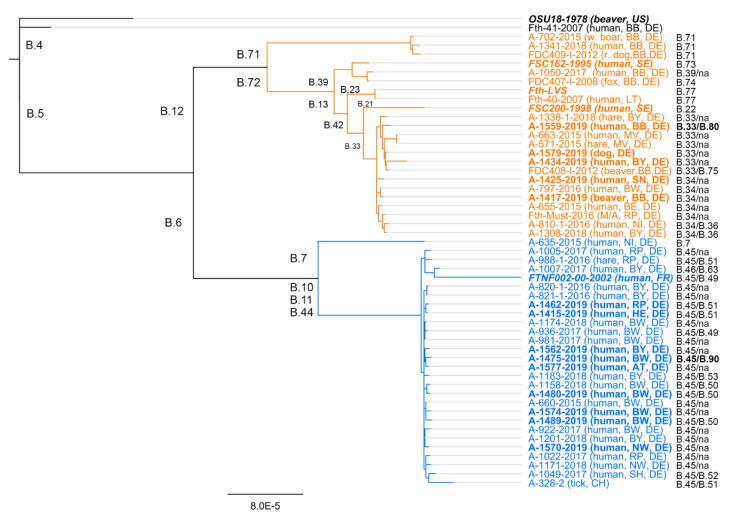
Phylogenetic relationship of *Francisella tularensis* subsp. *holarctica* isolates from 2007–2019 in Germany. The analysis was based on a Mauve alignment for collinear genomes. Genomes were generated by DNA sequencing and mapping of obtained DNA reads to the genome of *F. tularensis* subsp. *holarctica* (Fth) live vaccine strain (LVS) (for details, see [17]). For the clustering, the neighbor joining bootstrap method was chosen, with *F. tularensis* subsp. *holarctica* strain OSU18 as an out-group. Outlined for each genome are the identifier of the isolate and the year of sampling, the host organism, and the sampling spot (Germany’s federal state). In addition, the respective *Francisella* clade and final subclade (na = not determined) for each genome is given. The sequences obtained from isolates in 2019 (not published elsewhere yet) and some reference genomes included in the analysis are given in bold. Abbreviations: Austria (AT), France (FR), Lithuania (LT), United States (US), Sweden (SE) and Switzerland (CH). Germanys federal states: BB: Brandenburg; BE: Berlin; BW: Baden-Württemberg, BY: Bavaria; HE: Hesse, MV: Mecklenburg-Western Pomerania; NI: Lower Saxony; NW: North Rhine-Westphalia; RP: Rhineland-Palatinate; SH: Schleswig-Holstein; SN: Saxony; TH: Thuringia.

**Figure 2 microorganisms-08-01448-f002:**
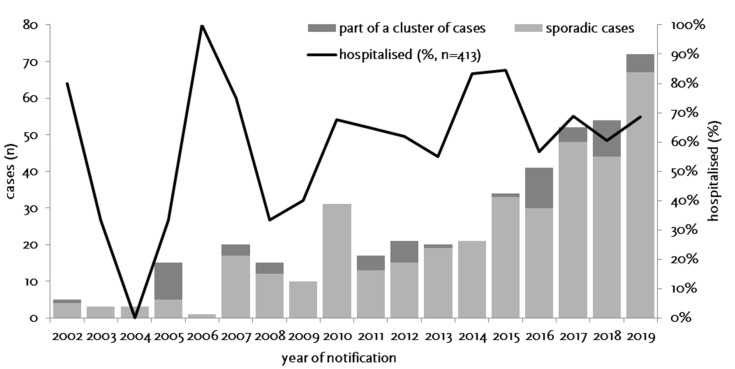
Notified cases of tularemia (sporadic and part of a cluster of cases, stacked bars) and proportion of hospitalized cases (line, second *y*-axis) by year of notification, Germany, 2002–2019.

**Figure 3 microorganisms-08-01448-f003:**
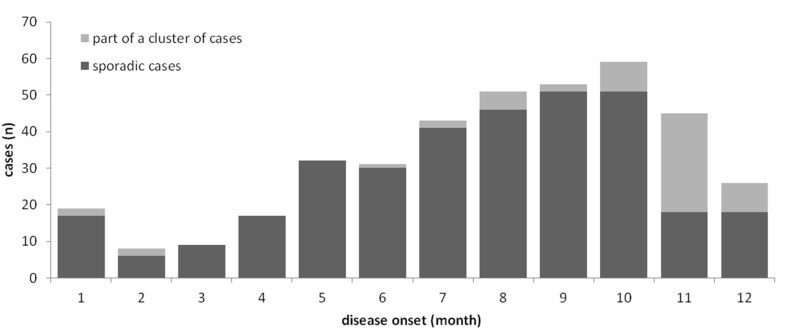
Notified cases of tularemia (sporadic and part of a cluster of cases, stacked bars) by month of notification, Germany, 2002–2019.

**Figure 4 microorganisms-08-01448-f004:**
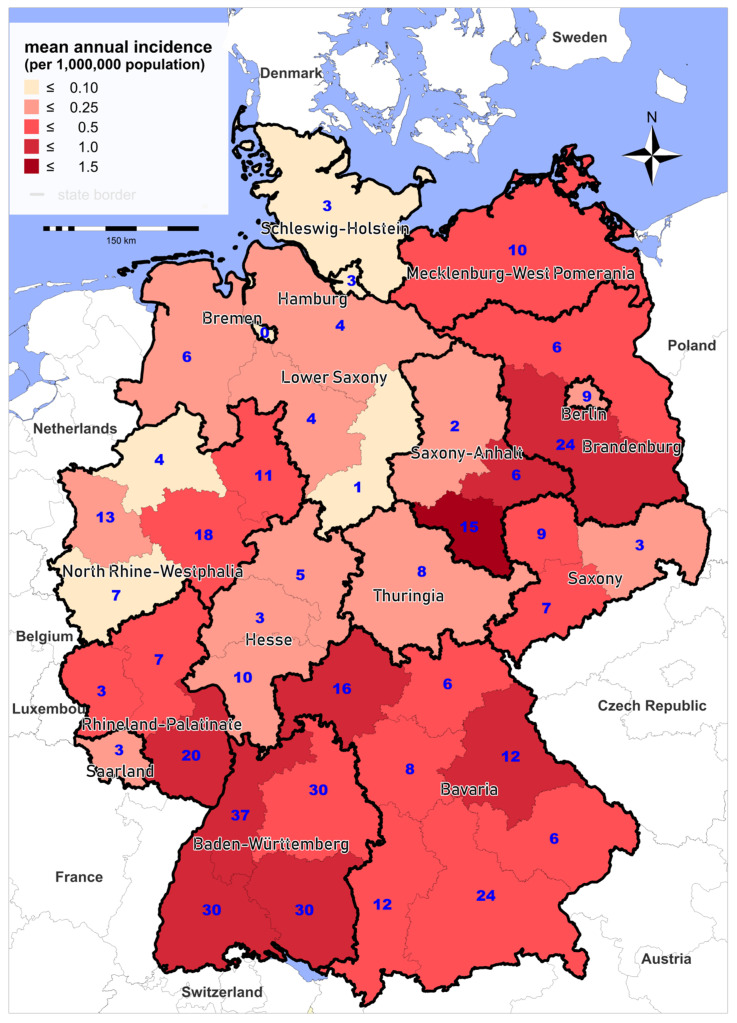
Mean annual incidence (per 1,000,000 population, shading) and total number of notified cases of tularemia (blue digits), Germany, 2002–2019.

**Table 1 microorganisms-08-01448-t001:** Notified tularemia cases with laboratory confirmation by clinical presentation, Germany, 2002–2019 (*n* = 435) [*n* = total number of tularemia cases].

Form	*n*	%
Glandular (lymphadenitis and not meeting criteria for other forms)	129	29.7
Ulceroglandular (lymphadenitis + skin ulcer)	68	15.6
Pneumonic (dyspnoea or pneumonia)	53	12.2
Intestinal (diarrhea, vomiting, or abdominal pain)	20	4.6
Oropharyngeal (lymphadenitis AND tonsillitis, pharyngitis, stomatitis)	23	5.3
Oculoglandular (lymphadenitis + conjunctivitis)	8	1.8
Combination (meeting criteria of >1 form)	52	12.0
Typhoidal	5	1.1
Other (symptoms not meeting any of the above criteria, e.g., “only fever”)	77	17.7

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
