# Peer review of "Francisella tularensis* Subspecies *holarctica* and Tularemia in Germany"

_microorganisms, 2020, doi:10.3390/microorganisms8091448_

Round 1
Reviewer 1 Report
The manuscript summarizes current knowledge on tularemia in Germany. The diversity and geographic spread of newly defined clades and subclades of F. tularensis subsp. holarctica in Germany are specified. The clinical and epidemiological aspects of tularemia cases that have occurred in Germany since 2002 are also described.
Comments.
Abstracts.
Line 17. Among arthropods, ticks are the primary vector of tularemia. Ticks are arachnids and not insects.
Line 18. I would not considered “food” as an “environmental source of contamination” but rather a source of foodborne tularemia.
Introduction
Line 37. “newt to arthropod vectors”. Does it mean arthropod bites ?
Line 38. “some times food” and drinking water.
Line 40. “human to human transmission has so far not been described”. See reference (Nelson CA, et al. Emerg Infect Dis. 2019;25(4):767-775) for human-to-human transmission of tularemia through organ transplantation.
Line 46. Some human tularemia cases have been recently described in Australia.
Paragraph 4. Tularemia in Germany
Line 150. “case definition applies to symptomatic persons”. It should be more accurate to specify here that case definition applies to “persons with clinical symptoms compatible with tularemia” together with one of the four laboratory criteria.
Paragraph 4.3. Clinical aspects and diagnosis
Line 253. The six major clinical forms of tularemia should be briefly described. In the same sentence the “typhoidal form” of tularemia is missing.
Line 269. “routing surveillance “ should be replaced by “routine surveillance)
Line 274. “Cultivation is exacerbated due to”. What is the meaning of exacerbated in this context? Does it mean poorly sensitive ?
Conclusion
Line 293. Please clarify “to be able to identify putative further (new) Francisella species.
Author Response
Reviewer 1:
The manuscript summarizes current knowledge on tularemia in Germany. The diversity and geographic spread of newly defined clades and subclades of F. tularensis subsp. holarctica in Germany are specified. The clinical and epidemiological aspects of tularemia cases that have occurred in Germany since 2002 are also described.
Comments.
Abstracts.
Line 17. Among arthropods, ticks are the primary vector of tularemia. Ticks are arachnids and not insects.
We deleted "insect bite" and introduced: "mosquitoes and ticks" (line 17)
Line 18. I would not considered “food” as an “environmental source of contamination” but rather a source of foodborne tularemia.
now: "... and by food." (line 18)
Introduction
Line 37. “newt to arthropod vectors”. Does it mean arthropod bites ?
Yes
Line 38. “some times food” and drinking water.
We introduced " and drinking water" as suggested by the reviewer (line 39)
Line 40. “human to human transmission has so far not been described”. See reference (Nelson CA, et al. Emerg Infect Dis. 2019;25(4):767-775) for human-to-human transmission of tularemia through organ transplantation.
We introduced the reference and now it is stated: "Human to human transmission has not been described so far, despite a case of tularemia after organ transplantation" (lines 41-42)
Line 46. Some human tularemia cases have been recently described in Australia.
This now is stated: " Recent reports of tularemia in Australia caused by F. tularensis subsp. tularensis demonstrated this subspecies to be present also in the Southern hemisphere" including two references (15, 16) (lines 53-55)
Paragraph 4. Tularemia in Germany
Line 150. “case definition applies to symptomatic persons”. It should be more accurate to specify here that case definition applies to “persons with clinical symptoms compatible with tularemia” together with one of the four laboratory criteria.
The sentence has been rephrased as suggested by the reviewer (lines 136-137)
Paragraph 4.3. Clinical aspects and diagnosis
Line 253. The six major clinical forms of tularemia should be briefly described. In the same sentence the “typhoidal form” of tularemia is missing.
We thank the reviewer for pointing this out and have added "typhoidal" to the list of clinical manifestations of tularemia. For a description of symptoms of the respective forms, we would like to refer the reader to table 1 or to WHO's Guidelines on Tularemia, which are referenced in the text. Furthermore, the review will be part of a special issue with probably further papers describing the clinical forms of tularemia in more detail. (line 250)
Line 269. “routing surveillance “ should be replaced by “routine surveillance)
It has been corrected (line 264)
Line 274. “Cultivation is exacerbated due to”. What is the meaning of exacerbated in this context? Does it mean poorly sensitive ?
Has been changed to: "poorly successful" (line 270)
Conclusion
Line 293. Please clarify “to be able to identify putative further (new) Francisella species.
The sentence has been rephrased : " The identification of the new Francisella species F-W12 in Germany indicated that it could make sense to have an 'open-view' (e.g. methods like Next Generation Sequencing [NGS]) when analyzing isolates or probes of patients suffering of tularemia to be able to identify further potentially existing (new) Francisella species in Germany. " (lines 324-327)
Reviewer 2 Report
This is a well written and comprehensive review of tularemia and Francisella species in Germany. I just have a few minor comments:
Line 81: Reword 'yet unknown to be present...' to 'have not previously been isolated from patients in Germany.'
Line 87-89: I suppose the question is what is the benefit to knowing that these isolates represent distinct 'sub-sub-sub clades' other than for classification purposes? For example, at this level is there any relationship with antimicrobial resistance or the what treatment is provided to patients?
Figure 1 legend, line 98: Were the isolates from 2019 shown in bold sequenced as part of this study (i.e. not published elsewhere)? If so, please state this. Otherwise what is the purpose of making these ones bold?
Line 114 (Francisella species F-W12 section): I find that having this section here disrupts the flow of the manuscript. I think this section would be better placed at the end of the manuscript.
Line 128: I don't understand the end of this sentence.
Line 241: Change 'get infected' to 'became infected'
Author Response
Reviewer 2
This is a well written and comprehensive review of tularemia and Francisella species in Germany. I just have a few minor comments:
We thank the reviewer for the kind words.
Line 81: Reword 'yet unknown to be present...' to 'have not previously been isolated from patients in Germany.'
It has been changed to "not previously been isolated" (line 92)
Line 87-89: I suppose the question is what is the benefit to knowing that these isolates represent distinct 'sub-sub-sub clades' other than for classification purposes? For example, at this level is there any relationship with antimicrobial resistance or the what treatment is provided to patients?
To our knowledge there is no relationship of such 'sub-sub-sub' clades to relevant phenotypes and often only one respective isolate is known.
Figure 1 legend, line 98: Were the isolates from 2019 shown in bold sequenced as part of this study (i.e. not published elsewhere)? If so, please state this. Otherwise what is the purpose of making these ones bold?
Yes, the isolates are part of this study and we now introduced: "(not published elsewhere yet)" (line 116)
Line 114 (Francisella species F-W12 section): I find that having this section here disrupts the flow of the manuscript. I think this section would be better placed at the end of the manuscript.
You are right and we now placed this part at the end of the manuscript (section 4. ) and we stated a cross reference at the introduction section: "This species seems not to be clinically relevant, regarding the yet available knowledge, and will be described in section 4. " (lines 65-66)
Line 128: I don't understand the end of this sentence.
It has been changed (deleted) (line 289)
Line 241: Change 'get infected' to 'became infected'
It has been corrected (line 234)